# Current Overview of Environmental Disinfection and Decolonization of *C. auris*: A Systematic Review from 2020 to 2025

**DOI:** 10.3390/tropicalmed10060155

**Published:** 2025-06-02

**Authors:** María Guadalupe Frías-De-León, Paola Betancourt-Cisneros, Erick Martínez-Herrera, Paola Berenice Zarate-Segura, Carlos Alberto Castro-Fuentes, Eduardo García-Salazar

**Affiliations:** 1Laboratorio de Micología Molecular, Unidad de Investigación Biomédica, Hospital Regional de Alta Especialidad de Ixtapaluca, Servicios de Salud del Instituto Mexicano de Seguro Social para el Bienestar (IMSS-BIENESTAR), Carretera Federal México-Puebla Km 34.5, Ixtapaluca CP 56530, Mexico; magpefrias@gmail.com (M.G.F.-D.-L.); paola14_02@hotmail.com (P.B.-C.); 2Sección de Estudios de Posgrado e Investigación, Escuela Superior de Medicina, Instituto Politécnico Nacional, Plan de San Luis y Díaz Mirón s/n, Col. Casco de Santo Tomas, Alcaldía Miguel Hidalgo, México City CP 11340, Mexico; erickmartinez_69@hotmail.com (E.M.-H.); pbzars@yahoo.com (P.B.Z.-S.); 3Unidad de Investigación, Hospital Regional de Alta Especialidad de Ixtapaluca, Servicios de Salud del Instituto Mexicano de Seguro Social Para el Bienestar (IMSS-BIENESTAR), Carretera Federal Mexico-Puebla Km 34.5, Ixtapaluca CP 56530, Mexico; castrofuenca@gmail.com

**Keywords:** propagation, far UV-C, outbreak control, candidiasis

## Abstract

*Candida auris* possesses distinctive features that facilitate its persistence and transmission in healthcare settings, causing outbreaks of infection that are difficult to treat. So, emphasis has been placed on implementing measures for controlling, eliminating, and preventing fungal transmission, such as environmental disinfection and patient decolonization. This review aimed to understand and analyze the agents for environmental disinfection and patient decolonization reported in the last 5 years. The PubMed database was reviewed, using the terms “*Candida auris*”, “disinfection”, and “decolonization”. Only original papers, published between 2020–2025, in English or Spanish, that included relevant information on the topic, were selected. After the selection process, 52 articles were chosen to analyze the agents for environmental disinfection and decolonization of *C. auris*. Natural and synthetic disinfectants and ultraviolet radiation were reported for the environmental disinfection, with variable efficacy, depending on factors such as concentration and exposure time. Natural and synthetic antiseptics were also reported for decolonization, with varying efficacy. For example, 2% chlorhexidine shows a 0.5 log reduction, while at concentrations >10% it is >4 log. However, most have only been tested in animal models. Based on the review, Far-UV-C radiation (222 nm) is safe and appropriate to mitigate (up to 1 log reduction) the spread of *C. auris* in the hospital setting. However, it is important to consider that the cost and limited availability of the device present a barrier to its implementation. Patient decolonization is still challenging nowadays due to the absence of agents with proven high efficacy in humans.

## 1. Introduction

The incidence of invasive fungal infections (IFIs) is increasing worldwide due to the growth of populations with typical risk characteristics for IFIs, which include prolonged intensive care unit (ICU) stays, use of mechanical ventilation or catheters, prolonged antibiotic treatment, and presence of underlying diseases (chronic kidney disease, diabetes mellitus, HIV infection, among others) [1]. In addition, new populations at risk have emerged, such as patients with COVID-19 who present frequent IFIs, particularly invasive candidiasis (IC) [1]. Several Candida species can cause IC, mainly *Candida albicans*, *C. parapsilosis*, *C. glabrata* (*Nakaseomyces glabratus*), and *C. auris* (*Candidozyma auris*) [2]. These yeasts were classified as high or critical priority fungal pathogens by the World Health Organization (WHO) [3], with *C. auris* standing out for causing severe IFIs with a high mortality rate worldwide (40–60%) [4]. Unlike the other species, *C. auris* is a non-commensal yeast, which is usually resistant to at least two of the main classes of systemic antifungals (93% are resistant to fluconazole, 35% to amphotericin B, and 7% to echinocandins) [5], limiting treatment options for patients [6]. In addition, this yeast has a halotolerant feature, which facilitates its transmission and causes outbreaks of infection that are difficult to treat in healthcare environments [7]. Due to its distinctive characteristics, *C. auris* represents a threat to public health worldwide.

Outbreaks of hospital-acquired infection have been associated with the ability of *C. auris* to colonize the skin, particularly areas exposed to high salinity and temperature, such as the armpit and groin [8], showing no signs of infection, and persisting for weeks [9]. Patients colonized by *C. auris* can develop invasive infections in 5–10% of cases, with high mortality rates [10]. Skin colonization facilitates the spread of this yeast to biotic surfaces (skin of healthcare workers and family members) and abiotic surfaces (mattresses, bed rails, thermometers, catheters, medical equipment, even to surfaces that are not in direct contact, such as floors, chairs, etc.) [11] of the hospital environment. They also persist for prolonged periods, resisting the action of disinfectants commonly used in hospitals [12]. It has been shown that the fungus can persist for up to 4 weeks on abiotic surfaces, remaining metabolically active, but not culturable, which makes detection difficult [13]. This explains how easily other patients can contract *C. auris* through contact with contaminated surfaces. For this reason, emphasis has been placed on implementing measures to control, eliminate, and prevent fungal transmission in the hospital environment. Some recommended measures are hygiene and disinfection of the patient’s environment. In the case of hygiene, it is critical to comply with the five moments for handwashing recommended by the WHO. In addition, it is necessary to follow some precautions, such as ensuring that infected and colonized patients are accommodated in isolated rooms, as well as cleaning these rooms daily. Regarding disinfection, it is crucial to thoroughly disinfect rooms at the time of patient discharge, using effective agents against microorganisms such as *Clostridium difficile* [9]. Another way to prevent and control outbreaks is by detecting patients colonized with *C. auris*, followed by their decolonization with topical antiseptics, such as chlorhexidine, since they can spread the fungus to other non-colonized patients. Decolonization and disinfection of areas during an outbreak has been shown to reduce the frequency of *C. auris* isolation in both clinical and colonization samples [14]. The rate of reduction in transmission risk associated with decolonization is rarely clearly known due to the influence of various factors, such as recolonization and the effectiveness of different interventions [14]. However, screening for this yeast at the time of patient admission or during their hospital stay is not routinely performed [15]. However, evidence justifies the need for screening, as patients with chronic respiratory diseases are at significant risk of persistent colonization [9]. It has even been shown that patients discharged from the hospital and reintegrated into the community usually take about 8 months to test negative for colonization [16]. Thus, controlling the spread of *C. auris* is still a challenge [17].

Due to the increasing incidence of *C. auris* infections and the various outbreaks that have occurred in several parts of the world [18], as well as the difficulty in controlling the spread of *C. auris*, it is essential to identify the current prevention and control strategies. Therefore, this study aimed to understand and analyze the agents for environmental disinfection and patient decolonization, which have been reported in the last 5 years.

## 2. Materials and Methods

A systematic literature search was conducted in four databases: PubMed, Scopus, SciELO, and EBSCO using the terms “*Candida auris*”, “disinfection”, and “decolonization”. For the systematic review, the PRISMA 2020 guidelines were used [19] (Figure 1). A protocol for this review was registered in PROSPERO’s international prospective register of systematic reviews (crd.york.ac.uk/prospero) with the ID CRD420251034737. The search was limited to articles published in English or Spanish from January 2020 to January 2025. After the automated database search and duplication were performed, three independent authors (M.G.F.D.-L., P.B.-C., and P.B.Z.-S.) determined the eligibility of articles based on title and abstract. The inclusion criteria were original articles, in English, on environmental disinfection strategies and decolonization of *C. auris*. The quality assessment for the risk of bias was performed in duplicate (E.M.-H., P.B.Z.-S., C.A.C.-F., and E.G.-S.) using two tools, the JBI Critical Appraisal Checklist for Systematic Reviews and Research Syntheses, which indicates in the general assessment that an article meets the necessary quality for publication, and the Critical Appraisal Skills Program (CASP), which establishes that a study has a logical development that makes it feasible for publication.

## 3. Results

After the selection process, 52 articles were chosen to analyze the strategies of environmental disinfection and decolonization of *C. auris*. Of these, 40 corresponded exclusively to strategies for disinfecting environments contaminated by *C. auris*, 11 to decolonization strategies, and one addressed both strategies.

The quality assessment of the studies using the JBI Critical Appraisal Checklist and CASP showed that 22 studies received a score of 10, 13 of nine, and 17 of eight. Based on the scores, 35 studies were classified as high quality and the remaining 17 as moderate methodological quality.

### 3.1. Disinfection of Environments Contaminated by C. auris

In the last 5 years, North America has been the region that has focused the most on improving and developing strategies for disinfecting the healthcare environment, followed by Europe, Asia, Africa, and South America (Table 1).

Various disinfectant agents for environmental disinfection, mainly for hard surfaces, have been reported in recent literature, including everything from natural and synthetic disinfectants to ultraviolet radiation (Table 1).

Among the natural disinfectants, lavender essential oil (*Lavandula angustifolia*), in free form or encapsulated in liposomes, has shown adequate results in eradicating *C. auris* in the form of primary and persistent biofilms on various surfaces [27]. In the same way that has been reported in other *Candida* species, the action of essential oils against *C. auris* entails producing reactive oxygen species (ROS) and affecting the expression of some biofilm-related genes [27].

Within the group of synthetic disinfectants, there is a wide range of agents, among which hydrogen peroxide, sodium hypochlorite, isopropyl alcohol, quaternary ammonium compounds, chlorhexidine, and benzalkonium chloride stand out due to the frequency of use (Table 1).

Hydrogen peroxide, both in 3.4 and 4.2% solution and aerosol, is effective in removing planktonic cells and *C. auris* biofilms on steel surfaces and medical equipment, without causing deterioration to surfaces, as it has relatively good compatibility with hard and soft surface materials [22,23,26,28,29,32,34,35,42]. The efficacy of hydrogen peroxide solutions at low concentrations, such as 1.7%, is limited [32], while the effectiveness of disinfection increases with the aerosol form, helping to stop nosocomial transmission of *C. auris* [29,34].

Sodium hypochlorite has variable efficacy, depending on the concentration. The load of *C. auris* on high-contact surfaces is considerably reduced with solutions of ≥ 1000 ppm and contact times greater than 1 min [34,46,51]. At concentrations ≥ 4000 ppm, *C. auris* removal is effective with only 1 min of contact [23]. The main disadvantage of using sodium hypochlorite is the incompatibility with some materials that constitute soft surfaces, such as mattresses, as it can lead to discoloration and deterioration [28].

Quaternary ammonium compounds have shown variable efficacy against *C. auris*. While some studies report that these compounds significantly limit fungal growth [25,32], others report that the biocidal effect of quaternary ammonium on *C. auris* is lower than that of other types of compounds, such as hydrogen peroxide, alcohol, or sodium hypochlorite [26,39,46,56].

Alcohol-based environmental disinfectants have shown satisfactory results in significantly reducing the load of *C. auris* on surfaces [26,32,40]. It has even been observed that *C. auris* is usually more susceptible to these disinfectants than *C. albicans* [25].

In recent years, reports on chlorhexidine as an environmental disinfectant have been scarce [45,46,48]. The few reports indicate that its effectiveness in eliminating *C. auris* on surfaces is variable, as it depends on the concentration and cleanliness of the environment [46]. The use of appropriate chlorhexidine concentrations leads to the elimination of 99.999% of *C. auris* biofilms [45], and its efficacy increases when coupled with silver sulfadiazine; nevertheless, it is still unknown whether the chlorhexidine–silver sulfadiazine binding has any clinical benefit [48].

Another environmental disinfectant that has been used is benzalkonium chloride; however, the outcome in all cases has been negative, as *C. auris* survived on surfaces, mainly wet wood [37,46]. Survival occurred due to the yeast’s efflux pump-mediated resistance [37].

Other synthetic disinfectants that have been evaluated for the elimination of *C. auris* in the hospital environment are peracetic acid, dodecylbenzenesulfonic acid, glutaraldehyde combined with quaternary ammonium and surfactant, potassium peroxymonosulfate, electrolyzed water, a combination of peracetic acid and hydrogen peroxide, furfuryl alcohol; 2-methyl-2-cyclopentenone; guaiac; potassium linoleate and silver nanoparticles, ozone [21,22,24,33,39,42,45,57]. All have shown fungicidal effects on *C. auris* on surfaces of different materials without causing deterioration. Furthermore, combining some of them, such as furfuryl alcohol, 2-methyl-2-cyclopentenone, and guaiac, increases the efficacy in removing preformed biofilms in stainless steel [57]. Potassium linoleate is a compound that has demonstrated biocidal activity against fungi and bacteria, and it can also be used in skin decolonization [33]. The silver nanoparticles of 1–3 nm in diameter have been evaluated on silicone elastomers and bandage fibers, demonstrating their fungicidal effect by inhibiting the biofilm formation (IC50 of 0.06 ppm) or destroying preformed ones (IC50 of 0.48 ppm). Silver nanoparticles are effective at low concentrations (0.06–0.48 ppm) and remain active even after repeatedly washing surfaces [24]. Another effective compound is ozone, which has eradicated *C. auris* from bed sheets after a 40-min exposure. The disadvantage of this compound lies in the resistance that some isolates have shown [21]. The use of substances that function as biofilm disruptors has also been reported, with high efficacy even in polymicrobial biofilms resistant to other disinfectants, such as chlorhexidine [45].

In the search for new compounds with fungicidal activity, a study evaluated 240 compounds, included in the Global Health Priority Box^®^, and found at least two compounds that are candidates for disinfectant development: hydramethylnon (MMV1577471) and flufenerim (MMV1794206). These compounds inhibit the growth of *C. auris* by more than 58% [55].

Ultraviolet radiation, type A (UV-A), B (UV-B), and C (UV-C), has emerged as a support method to traditional chemical disinfection methods. The least commonly used type of radiation is UV-A, perhaps because, although *C. auris* is photosensitive to different wavelengths, it is not as sensitive as other yeasts (*Saccharomyces cerevisiae*) [47]. Short-wave UV-B, at a dose of 51.3 mJ/cm^2^, has also achieved inactivation of *C. auris* in solutions [60]; however, no other recent studies support its application. UV-C is the most studied type of radiation for disinfection [49]. Exposure to UV-C (222–280 nm), at distances of 1–3 m and times of 1–45 min, results in reducing the load of *C. auris* and its biofilms on various surfaces (bed sheets, stainless steel, polystyrene, fabrics, glass) [30,35,36,41,50,53,60]. However, a disadvantage of using UV-C is that the sensitivity of *C. auris* depends on the clade to which they belong, i.e., isolates from clades I, II, and IV are more sensitive than those from clade III [20], so the efficacy of UV-C can vary between geographical regions.

Far UV-C exposure in the range of 200–230 nm for 45 to 120 min is effective in reducing the load of *C. auris* on steel surfaces, wheelchairs, portable equipment in clinical areas, etc. [49,54,55]. Disinfection by this method is safe because it cannot penetrate the stratum corneum of the human skin [60]. However, it is important to mention that there are factors that can reduce its effectiveness and, therefore, limit its use. These factors include surface characteristics (porosity, presence of residue or dirt), exposure time, and the distance between the UV-C source and the contaminated surface [50]. One limitation is that it can interact with atmospheric molecules such as oxygen and volatile organic compounds, forming ozone and harmful radicals indoors.

Ultraviolet germicidal radiation (UVGI) is a method that inactivates *C. auris* in aqueous solutions with an efficacy of 99.999%, using a dose of 192 mJ/cm^2^ [52].

Notably, antimicrobial surfaces have also been developed to reduce the load of *C. auris* to the minimum limit of detection [31,44]. These surfaces comprise a self-disinfecting polymer that generates a surface layer of acidic water when hydrated, which causes damage to the microorganisms that come into contact with it [44]. However, more research is required on the mechanism of action of these surfaces to apply them in clinical settings [31].

### 3.2. Decolonization of C. auris

One of the most challenging issues while controlling the spread of *C. auris* in healthcare settings has been decolonization, i.e., reducing or eliminating the fungal load in a patient’s body. The evaluation or development of new strategies to decolonize *C. auris* has been conducted mainly in North America, Asia, and Europe (Table 2).

Chlorhexidine is the most widely used synthetic antiseptic [62,66,67,71]; however, the reduction in fungal load is slight, so it can sometimes present clinical failures [63]. It has been reported that chlorhexidine activity against *C. auris* in the skin can be enhanced with isopropanol, and the decolonizing effect of chlorhexidine/isopropanol is further enhanced by the combined use of natural antiseptics, such as tea tree and lemongrass oil [63]. In order to increase the effectiveness of chlorhexidine, a formulation based on the proprietary Advanced Performance Technology (APT™) platform was proposed [65]. This formula combines FDA-approved inactive ingredients (ascorbic acid, carbomer, cholecalciferol, citric acid, diazolidinyl urea, dimethyl sulfoxide (DMSO), dipropylene glycol, glycerin, polysorbate 20, sodium dodecylbenzenesulfonate, tetrasodium EDTA, tocopheryl acetate (vitamin E acetate), triethanolamine, water, and aloe barbadensis leaf juice) with an active pharmaceutical ingredient (chlorhexidine) to effectively reduce the burden of *C. auris* in mouse skin tissue. However, its efficacy in humans needs to be evaluated. Independently, other natural (manuka oil) and synthetic antiseptics (povidone iodine, nystatin, quaternary ammonium iodine tincture, and 75% ethanol, chlorine) have shown fungicidal activity against *C. auris*, with products containing iodine and benzalkonium chloride being the most effective *in vitro* [21,68,71]. Some compounds with antimicrobial activity, such as triclosan, boric acid, and zinc oxide, which are primarily used in personal care products, have also been tested. Boric acid triclosan showed antifungal activity against *C. auris*, but zinc oxide did not [70]. So, further research is needed to determine whether these compounds can reduce the burden of *Candida* on skin.

## 4. Discussion

Since its emergence in 2009, *C. auris* has posed a global health threat, partly due to the ease with which it spreads and persists in the hospital environment. For this reason, prevention and control protocols have been established, in which environmental disinfection and decolonization are essential. There are recommendations by various competent organizations, such as the Centers for Disease Control and Prevention, and WHO, to reduce nosocomial transmission of *C. auris* [72]. However, the increase in infection outbreaks since the COVID-19 pandemic has prompted the study of the efficacy of commonly used disinfectant agents, as well as the search for new, more effective agents for environmental disinfection and patient decolonization.

In environmental disinfection, the findings of the latest studies highlight the need to choose the right agent since not all commonly used disinfectants are effective against *C. auris*. For example, benzalkonium chloride, which is widely used in hospitals to disinfect hands and devices, does not have an adequate effect on *C. auris*, as the fungus appears to develop resistance [37]. Sodium hypochlorite-based disinfectants, which are also frequently used in healthcare settings, have a biocidal effect on *C. auris* when used at high concentrations, causing oxidative damage to the cell membrane and essential intracellular components [73]. However, at concentrations <4000 ppm, a prolonged contact time is required to achieve the same effect [23]. This can be impractical in hospitals, as it can hinder the dynamics of patient care.

Given the drawbacks of commonly used environmental disinfectants, the need for effective agents against *C. auris* has led to the evaluation of various compounds, some very simple, such as essential oils, which could be integrated into existing disinfectant formulations to achieve better results due to their effectiveness. Alternatively, there are more complex ones such as silver sulfadiazine combined with chlorhexidine, which has a clear inhibitory effect on *C. auris*
*in vitro*, but whose clinical benefit has not been confirmed [48]. Hydramethylnon and flufenerim are two compounds in which activity against *C. auris* has been discovered [55]. However, it still needs to be determined whether they can penetrate the protective layers of biofilms and eliminate viable *C. auris* cells before using them to develop new disinfectants.

One of the most interesting advances in the attempt to mitigate the persistence of *C. auris* in the healthcare environment is the development of antimicrobial surfaces, which destroy microorganisms that come into contact with them [44]. It would be ideal if the surfaces in contact with patients were covered by the polymers that constitute these antimicrobial surfaces. However, studies on this topic are still required.

It is worth noting that an innovative and safe approach to air and surface disinfection is far UV-C. This type of radiation can be applied in rooms, even when people are present, since UV-C cannot penetrate even the most superficial layer of the skin [60]. It is worth noting that the studies report that Far UV-C (222-nm) was evaluated rather than 254-nm UV-C due to safety considerations. Far UV-C doses within threshold limit values proposed by the American Conference of Governmental Industrial Hygienists (ACGIH) and the International Commission on Non-Ionizing Radiation Protection (ICNIRP) may be safe in occupied areas. Thus, accidental exposure to far UV-C, but not 254-nm UV-C, would pose minimal risk [49]. However, it is important to consider the limited availability of amplifiers and accessories due to their relatively new technology; the higher initial costs of lamps and accessories compared to mercury vapor lamps; high energy consumption; limited data on potential long-term health effects; and ozone production are some barriers to their implementation.

Patient decolonization continues to represent a challenge in controlling the spread of *C. auris* since few studies focus on the subject. Chlorhexidine, widely used in hospitals, has limited activity against *C. auris*, which has led to failure rates ranging from 76.3–81.2% [63,66]. New chlorhexidine formulations have been designed to improve their efficacy. However, they have only been used in animal models, and the efficacy in humans has yet to be evaluated.

Finally, it is necessary to acknowledge that this review has an important limitation, which is the linguistic bias, since only publications in English or Spanish were considered. This exclusion may lead to the omission of relevant data in other languages and alter the conclusions of the review.

## 5. Conclusions

Recent studies on disinfectant and decolonizing agents for *C. auris* demonstrate the importance of choosing the right agent, since there may be different susceptibility between isolates of different clades. In addition, not all existing disinfectants show adequate activity against *C. auris*. One of the most promising strategies to mitigate the spread of *C. auris* is the use of far-UV-C radiation, as it is safe for the patient and healthcare staff and has the potential to disinfect both surfaces and air in the hospital environment effectively. However, it is important to consider that the cost and limited availability of the device present a barrier to its implementation.

Lastly, patient decolonization remains a challenge nowadays. Some disinfectants have shown promising results in animal models, but the effect on human skin has not been studied yet.

## Figures and Tables

**Figure 1 tropicalmed-10-00155-f001:**
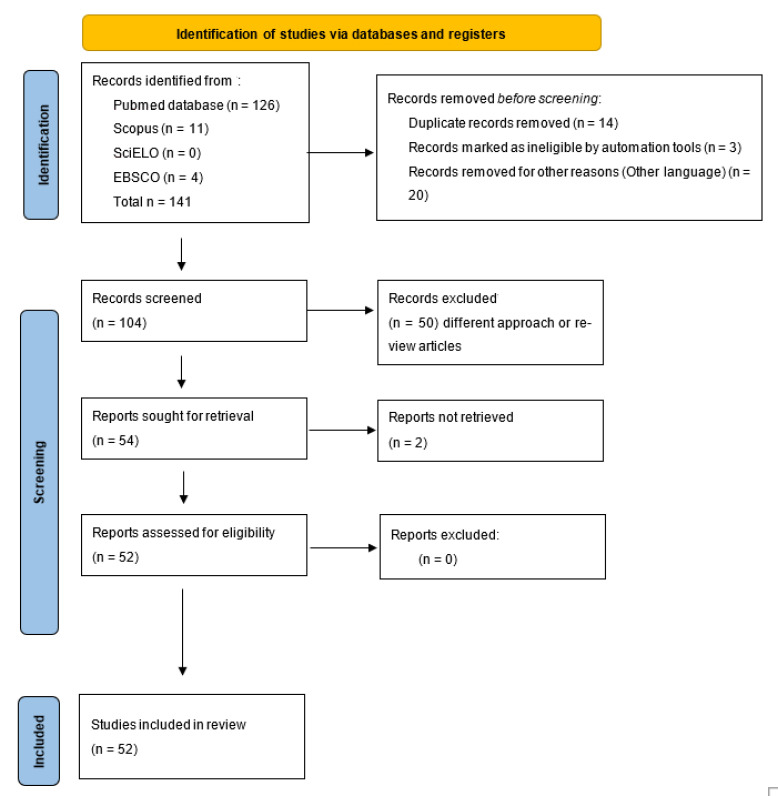
Prisma 2020 flowchart of the data extracted for the systematic review from bibliographic search.

**Table 1 tropicalmed-10-00155-t001:** Disinfection agents for the hospital environment contaminated with *Candida auris.*

Agent	Concentration or Wavelength	Exposure Time	Logarithmic Reduction	Strain/Clade	Country	Reference
UV-C on steel surfaces	200–280 nm	30 min at 1.5 m distance	0.52 log_10_ 1.38 log_10_ 0.04 log_10_ 1.15 log_10_	I II III IV	USA	[20]
UV-C and ozone on bed sheets	UV-C 253.7 nm Ozone 300 mg/m^3^	60 min, at 2.5 m distance 20 min, two cycles	3.22 log_10_ 3.260 log_10_	III	China	[21]
C_2_H_4_O_3_ and H_2_O_2_ on portable medical equipment surfaces	22% H_2_O_2_ and 4.5% C_2_H_4_O_3_	A 21-min cycle	*C. auris* was eradicated	I	USA	[22]
Two chlorine-based disinfectants	500, 1000, 2000, 4000 and 6500 ppm	1 min of wet contact	>3 log_10_ to ≥4000 ppm	I	USA	[23]
AgNP (1 to 3 nm in diameter) in silicone elastomer and bandage fibers	2.3 to 0.017 ppm	24 h	5.7 log_10_	I	USA	[24]
Two surface disinfectants, an alcohol-based disinfectant and a QAC-based disinfectant	Alcohol 35% QAC 0.25%	0.5 min contact 1 min contact	NR	DSM-21092	Germany	[25]
Five H_2_O_2_ or alcohol-based disinfectants and four QAC-based disinfectants	H_2_O_2_ 0.5 and 1.4% QAC 0.084, 0.5, 0.61, 2, 10.9 y 21.7%	H_2_O_2_ 1, 3 and 10 min QAC 1, 2 and 10 min	≥5.32 log_10_ 0.25 to ≥5.29 log_10_	IV	USA	[26]
*Lavandula angustifolia* essential oil, in free form or encapsulated in liposomes	0.005–0.5% *v*/*v*	24 and 48 h	*C. auris* was eradicated	DSM 21092	Italy	[27]
H_2_O_2_-based disinfectant, and NaClO-based disinfectant	>0.1–4% H_2_O_2_ 0.65% NaClO	1 min 1 min	≥5.1 log_10_ ≥6.1 log_10_	II (AR-0381) and IV (AR-0385)	USA	[28]
Accelerated H_2_O_2_ for environmental disinfection twice a day	NR	Twice a day	NR	I	Canada	[29]
Robotic UV-C	254 nm, 2.7 mJ/cm^2^ per second	20–25 min at 1 m distance	>1.7 log_10_	NCPF 8971, NCPF 8977, NCPF 8984 and DSM 21092	Austria	[30]
CSC	97.5–100% sodium chloride compressed into blocks	1 min	2.15 log_10_	UAMH 12148	Canada	[31]
ETH QAC ALD PP H_2_O_2_ mH_2_O_2_	100% 100% 0.5% 3% 3.4% 4.2% 5% 10%	1 min 1 min 0.5% 30 min 5 min 5 min 15 min 15 min	>5 log_10_ >5 log_10_ >5 log_10_ >7 log_10_ >2 log_10_ >3 log_10_ No activity >2 log_10_	NCPF8971, NCPF8977, NCPF8984, DSM21092	Austria	[32]
C18H31KO2 (isomerized)	86 mM	48 h	NR	NR	USA	[33]
Chlorine solution. and a H_2_O_2_ nebulization	≥1000 ppm on high-contact surfaces 5000 ppm at patient discharge	NR	NR	I	Italy	[34]
UV-C aHP	254 nm, 900 mW/cm^2^ at 1 m and 450 mW/cm^2^ at 2 m 6% stabilized with silver	20 min 1 h	*C. auris* was eradicated	III	South Africa	[35]
UV-C on stainless steel, plastic/polystyrene, and polycotton fabric surfaces	252–280 nm, 5, 10, 20 and 40 mJ cm^−2^	5 s 10 s 20 s 40 s	2.9 log_10_	II	USA	[36]
Sodium dichloroisocyanurate (NADCC)Ethanol Benzalkonium chloride (BC) H_2_O_2_ on different surfaces	1000 ppm 70% 95% NR	24 h	C. *auris* was eradicated 4.36 log_10_ *C. auris* was eradicated 5.4 log_10_	Cau 4888, Cau 3499 and Cau 6326	South Africa	[37]
One-step anionic surfactant disinfectant (active ingredient C18H30O3S)	0.29%	1 min	5.64 log_10_ (clade I) 5.2 log_10_ (clade II) 4.97 log_10_ (clade III) 4.78 log_10_ (clade IV)	I (AR-0389) II (AR-0381) III (AR-0383) IV (AR-0385)	USA	[38]
23 liquid disinfectants	QAC-Alcohol 0.25% H_2_O_2_ 0.5%	NR	NR	I, II, III and IV	USA	[39]
Quaternary ammonium and isopropyl alcohol-based germicidal wipe (Sani-Cloth^®^, PDI Healthcare, Woodcliff Lake, NJ, USA) on medical equipment surfaces, and 0.65% sodium hypochlorite on high-contact surfaces	sodium hypochlorite 0.65%	NR	NR	III	USA	[40]
UV-C on contact surfaces (steel, plastic, and glass supports/holders) in laboratory and hospital environments	254 nm	10 min	2.93 log_10_	I	Poland	[41]
EW, NaDCC, and PAA/H_2_O_2_ applied by electrostatic sprayers	NR >4000 ppm 2000 ppm/0.5%	1 min	1.57 log_10_ 1.15 log_10_ 1.26 log_10_	I	USA	[42]
H_2_O_2_ QAC-isopropyl alcohol based disinfectant wipes (EPA approved)	0.5% 0.25–55%	1, 2, 3 and 10 min	>6 log_10_	MYA-5001	USA	[43]
Solid surface of a self-disinfecting anionic block polymer that inherently generates a surface layer of acidic water when hydrated (pH < 1)	52 mol% midblock sulfonation	At contact, after hydration	NR	I	USA	[44]
CHD and BD (BlastX, Torrent, NSSD) on surfaces	NR	NR	NR	I	USA	[45]
Chlorine chlorhexidine benzalkonium chloride	200 ppm 500 ppm 2 and 4% NR	1, 5 and 30 min	>3 log_10_ (clade I) with 500 ppm >3 log_10_ with 4%Resistance	I and IV	Turkey	[46]
Photoinactivation with UV-A	365, 400, and 450 nm	NR	1 log_10_	I	Germany	[45]
Far UV-C UV-C short-wave UV-B on yeast solutions	222 nm 254 nm 302 nm	4.3 mJ/cm^2^ 6.1 mJ/cm^2^ 51.3 mJ/cm^2^	1 log_10_	II	Germany	[47]
CHD and CHD-S impregnated in segments of central venous catheters	0.03 to 512 μg/mL	24 h	NR	II, IV	USA	[48]
Far UV-C on bathroom surfaces	222 nm	2 h, 11.7 µW/cm^2^ in direct line and 0.4 µW/cm^2^ in non-direct line	≥1.2 log_10_	AR0385 (Clade IV)	USA	[49]
UV-C on hard surfaces	254 nm	250 mJ/cm^2^ for 7 min at 2.4 m distance	≥3.86 log_10_	AR0385 (IV)	USA	[50]
Ten-fold diluted NaClO in medical devices	10%	NR (intensive)	*C. auris* was eradicated	II	Korea	[51]
UVGI to inactivate *C. auris* strains in aqueous solution	254 nm	10, 20, 30, 40, 50, 60, 70, 80, 90, 100 and 150 mJ/cm^2^ at a 27.9 cm distance	5 log_10_ at a dose of 66 to 110 mJ/cm^2^	I, II, III and IV	USA	[52]
UV-C	254 nm	596.62 ± 27.98 mJ/cm^2^, 2.74 m	>6 log_10_	ATCC MYA-5001	USA	[53]
Far UV-C installed on the wall.	254 nm	45 min at a 2 and 3 m distance	<3 log_10_	I	USA	[54]
240 compounds from the Global Health Priority Box^®^	Hydramethylnon (MMV1577471) 16 μg/mL Flufenerim (MMV1794206) 4 μg/mL	NR	NR	IV	Brazil	[55]
Detergent, microfiber mop UV-C Disinfectant with non-sporicidal activity based on QAC Disinfectant with sporicidal activity based on NaClO	UV-C 252 nm QAC NaClO 0.25%	NR	NR	I	USA	[56]
C5H6O2, C_6_H_8_O and C7H8O2 on stainless steel	8% *v*/*v* 9% *v*/*v* 2% *v*/*v*	24 h 24 h 24 h	6.3 log_10_	I	India	[57]
Far UV-C on the surface of portable equipment in clinical areas	222 nm	4 and 12 h	>2 log_10_ after 4 h >3 log_10_ after 12 h	I	USA	[58]
Sodium hypochlorite	0.5 at 1%	Three times a day and after patient’s discharge	NR	I	India	[59]

UV-C: Ultraviolet light type C; AgNP: Silver nanoparticles; C_2_H_4_O_3_: Peracetic acid; H_2_O_2_: Hydrogen peroxide; C18H30O3S: Dodecylbenzenesulfonic acid; ETH: Ethanol-based disinfectants; QAC: Quaternary ammonium; ALD: Combination of glutaraldehyde, quaternary ammonium and surfactant; PP: potassium peroxymonosulfate; mH_2_O_2_: micellar formulation with 17% *v/v* H_2_O_2_; C_5_H_2_O_2_: furfuryl alcohol; C_6_H_8_O: 2-methyl-2-cyclopentenone; C7H_8_O_2_: guaiacol; MIC: minimum inhibitory concentration; EPA: United States Environmental Protection Agency; CHD: Chlorhexidine; BD: Biofilm disruptors; ROS: Reactive oxygen species; NaClO: Sodium hypochlorite; ICU: Intensive care unit; CHD-S: Chlorhexidine silver sulfadiazine; UVGI: Ultraviolet germicidal irradiation; CSC: compressed sodium chloride antimicrobial surface; EW: electrolyzed water; NaDCC: sodium dichloroisocyanurate; PAA/H_2_O_2_: peracetic acid/hydrogen peroxide; aHP: aerosolized H_2_O_2_; C18H_2_KO_2_: potassium linoleate. NR: Not reported.

**Table 2 tropicalmed-10-00155-t002:** Decolonization agents for *Candida auris.*

Decolonization Agent	Concentration	Exposure Time	Logarithmic Reduction	Strain/Clade	Country	Reference
Disinfectant 84 comprises chlorine iodine tincture QAC 75% ethanol benzalkonium bromide	1000 mg/L 2% (*w*/*v*) 2000 ppm 75% (*v*/*v*) 1000 mg/L	1 min 1 min 10 min 15 s 10 min	3 log_10_ NR Resistance 3 log_10_ NR	CBS12766, INCa-1, INCa-2	China	[21]
Iodine, silver, polyhexamethylene biguanide, octenidine, hypochlorous acid, benzalkonium chloride, surfactant-based topical containing poloxamer 188	NR	24 h	>6 log_10_ >6 log_10_ >6 log_10_ Resistance <1 log_10_ >6 log_10_ 1 log_10_	B11903	USA	[61]
CHX	2%	1 s	Complete eradication of skin colonization	I, II, III, IV, NIH, MYA-2876	USA	[62]
CHX Isopropanol tea tree oil (*Melaleuca alternifolia*) lemongrass oil (*Cymbopogon flexuosus*)	2% 70% 10% 5%	1 h daily for 3 days	0.5 log_10_ 1 log_10_ 1.5 log_10_ NR	I	USA	[63]
HA (isolated from *Shiraia bambusicola* and *Hypocrella bambusae*) bound to a novel organic compound (COP1T) with PEG chains	0.78 µg/mL 1.56 µg/mL 3.125 µg/mL	30 °C under a 470 nm laser (MDL-III-470 nm, 100 mW/cm^2^) for 30 min	4.2 log_10_ 4.1 log_10_ 2.7 log_10_	BJCA001	China	[64]
CHG in Advanced Penetration Technology (APT™) formulation	CHG 3.39% with APT	twice daily for 7 days	NR	I	USA	[65]
CHX	4%	daily bath for 1 week	NR	I	Bahrain	[66]
Wash mitts impregnated with CHG or OCT-based antiseptic	97% ≥10%	30 s	3 log_10_ ≥4 log_10_	DSM 21092, DSM 105986	Germany	[67]
Sodium hypochlorite isotonic solution	0.1%	daily bath	NR	I	USA	[65]
Synthetic (CHX, povidone iodine, and nystatin) natural (tea tree and manuka oil) antiseptics	NR <1.25% (*v*/*v*)	daily bath, contact with wounds	NR	III	China	[68]
PG and CAP Ointment	1 and 0.8%	3 h	5 log_10_	0391	USA	[69]
Triclosan Boric acid Zinc oxide	0.2 and 0.3% 1.9 and 5.0% 8.6 and 25%	48 h	NR	IV	Colombia	[70]

CHG: chlorhexidine digluconate; OCT: octenidine dihydrochloride; CHX: chlorhexidine; QAC: quaternary ammonium; HA: hypocrelin A; PEG: polyethylene glycol; PG: polygalacturonic acid; CAP: caprylic acid.

## Data Availability

The original contributions presented in this study are included in the article. Further inquiries can be directed to the corresponding author.

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
