# Peer review of "Current Overview of Environmental Disinfection and Decolonization of C. auris: A Systematic Review from 2020 to 2025"

_tropicalmed, 2025, doi:10.3390/tropicalmed10060155_

Round 1
Reviewer 1 Report
Comments and Suggestions for Authors
although many reviews of the literature on C.auris are now available, the review on the use of disinfectants and decontaminants presented by the authors is of interest to readers. the work is well written, well presented and a smooth read.
it is emphasised that C.auris and Candida spp are always to reporte in italics
Author Response
REVIEWER 1
Although many reviews of the literature on C. auris are now available, the review on the use of disinfectants and decontaminants presented by the authors is of interest to readers. The work is well written, well presented and a smooth read.
it is emphasised that C. auris and Candida spp. are always to reporte in italics
Answer: Thank you for your feedback, we have reviewed the entire document and italicized all scientific names.
Reviewer 2 Report
Comments and Suggestions for Authors
Review of Manuscript: "Current Overview of Environmental Disinfection and Decolonization of C. auris: A Systematic Review"
The manuscript by De Leon et al. provides a comprehensive review of disinfection and decolonization strategies for Candida auris, a critical emerging pathogen. However, the following comments need be addressed:
Title: I suggest that the date of studies reviewed should reflect on the tile e.g. 2020–2025
24: This phrase "variable efficacy, depending on factors such as concentration, and exposure time" should be clarified by stating the concentration of the chemical agent that was effective and ineffective, and the log reduction achieved
26: The conclusion about far-UV-C safety lacks context. What is the wavelength of the UV and the log of reduction of contamination on different surfaces without huma toxicity?
Introduction
42-44: Quantify the C. auris resistance to antifungals e.g. 90% resistance rate to fluconazole and 40% to amphotericin B
68-70: How is the decolonization done? Tie the rationale for decolonization e.g. what rate does the decolonization reduce transmission?
Methodology
Why was it only Pubmed that was searched? Why were other databases not searched?
Was the review registered in PROSPERO or other databases?
The PRISMA flowchart (Figure 1) is not showing the complete texts in the boxes, and it lacks detail on exclusion reasons
Risk of bias assessment is mentioned but not detailed e.g., no summary table or scores
Results
Table 1: Some entries lack key details e.g., strain clades, sample size
The Efficacy metrics (e.g., log reduction) are absent for many agents.
Authors should have standardize the columns e.g., Agent, Concentration, Exposure Time, Log Reduction, Strain/Clade, Reference
Consistent unit(e.g., ppm, log10) should be used I the table
165: This phrase “both in 3.4 - 4% solution” should be clarified because this is a range
209: There should be quantitative support for this claim "silver nanoparticles inhibited biofilms"
Discussion
The authors overstates UV-C applicability without noting limitations
There should be citation for failure rates for decolonization
What is/are the limitations of the review
Conclusions
The conclusion on far-UV-C omits implementation barriers such as the cost and device availability
Italicize all the names of organisms including botanical names of plants
Author Response
REVIEWER 2
Review of Manuscript: "Current Overview of Environmental Disinfection and Decolonization of C. auris: A Systematic Review"
The manuscript by De Leon et al. provides a comprehensive review of disinfection and decolonization strategies for Candida auris, a critical emerging pathogen. However, the following comments need be addressed:
Title: I suggest that the date of studies reviewed should reflect on the tile e.g. 2020–2025
Answer: We appreciate the suggestion and agree, the title has been changed.
24: This phrase "variable efficacy, depending on factors such as concentration, and exposure time" should be clarified by stating the concentration of the chemical agent that was effective and ineffective, and the log reduction achieved
Answer: The sentence was clarified by giving an example, since the results are very diverse. All details can be reviewed in the manuscript content.
26: The conclusion about far-UV-C safety lacks context. What is the wavelength of the UV and the log of reduction of contamination on different surfaces without huma toxicity?
Answer: The information was supplemented. It is worth noting that the studies report that Far UV-C (222-nm) was evaluated rather than 254-nm UV-C due to safety considerations. Far UV-C doses within threshold limit values proposed by the American Conference of Governmental Industrial Hygienists (ACGIH) and the International Commission on Non-Ionizing Radiation Protection (ICNIRP) may be safe in occupied areas. Thus, accidental exposure to far UV-C, but not 254-nm UV-C, would pose minimal risk (Kaple et al., 2024).
Introduction
42-44: Quantify the C. auris resistance to antifungals e.g. 90% resistance rate to fluconazole and 40% to amphotericin B
Answer: Resistance percentages were included, according to the literature.
68-70: How is the decolonization done? Tie the rationale for decolonization e.g. what rate does the decolonization reduce transmission?
Answer: Decolonization is performed with topical antiseptics, the most common being chlorhexidine, which help reduce the fungal load on the skin. This measure helps reduce the risk of transmission, but we do not have information regarding a specific percentage of this risk reduction, as it has been reported that it is difficult to determine due to the influence of various factors, such as recolonization and the effectiveness of the different interventions. This information was included in the manuscript.
Methodology
Why was it only Pubmed that was searched? Why were other databases not searched?
Answer: The search was expanded in Scientific Electronic Library Online (SciELO), SCOPUS y EBSCO.
Was the review registered in PROSPERO or other databases?
Answer: Registration was carried out in PROSPERO. This information was included in Materials and Methods.
The PRISMA flowchart (Figure 1) is not showing the complete texts in the boxes, and it lacks detail on exclusion reasons
Answer: Excuse us. Figure 1 has been adjusted to display the full text. The reasons for excluding some articles are mentioned.
Risk of bias assessment is mentioned but not detailed e.g., no summary table or scores
Answer: The scores were included in the new version of the manuscript.
Results
Table 1: Some entries lack key details e.g., strain clades, sample size
The Efficacy metrics (e.g., log reduction) are absent for many agents.
Authors should have standardize the columns e.g., Agent, Concentration, Exposure Time, Log Reduction, Strain/Clade, Reference
Consistent unit(e.g., ppm, log10) should be used I the table
Answer: Thank you for your feedback, the Tables were modified.
165: This phrase “both in 3.4 - 4% solution” should be clarified because this is a range
Answer: You are right, the sentence was clarified.
209: There should be quantitative support for this claim "silver nanoparticles inhibited biofilms"
Answer: The concentration data was included.
Discussion
The authors overstates UV-C applicability without noting limitations
Answer: We regret this interpretation and appreciate the opportunity to correct it. Honestly, we were only trying to interpret what we found in the review; we never had any interest in favoring or exaggerating any one disinfection method. But we fully agree that, although UV-C is effective, it has significant limitations, which we have included in this new version of the manuscript.
There should be citation for failure rates for decolonization
Answer: The decolonization failure rate was included.
What is/are the limitations of the review
Answer: Limitations of the study were included.
Conclusions
The conclusion on far-UV-C omits implementation barriers such as the cost and device availability
Answer: The conclusion was modified.
Italicize all the names of organisms including botanical names of plants
Answer: It was written in italics.
Reviewer 3 Report
Comments and Suggestions for Authors
Dear Authors,
your review is well-written and comprehensively analyzes the bibliography of the last five years regarding decolonization and disinfection of Candida auris.
I would like to offer a few minor suggestions that, in my opinion, could further enhance your work.
L 29. To optimize keyword effectiveness, I recommend removing 'C. auris', ‘environmental disinfection’ and ‘decolonization’ since keywords should ideally be unique terms not found in the title.
L 52. High mortality rates
L 56. I suggest considering 'commonly' or 'routinely' as more precise alternatives to 'frequently'.
L 90, 156, 158, 170, 174, 178, 180, 184, 185, 187, 190, 194, 197, 202, 212, 223, 225, 228, 230, 235, 238, 240, 247, 249. C. auris (italics)
L 155. Lavandula angustifolia (italics)
L 185. C. albicans (italics)
L 271. In vitro (italics)
Author Response
Reviewer 3
Dear Authors,
your review is well-written and comprehensively analyzes the bibliography of the last five years regarding decolonization and disinfection of Candida auris.
I would like to offer a few minor suggestions that, in my opinion, could further enhance your work.
Answer: We welcome and appreciate your feedback.
L 29. To optimize keyword effectiveness, I recommend removing 'C. auris', ‘environmental disinfection’ and ‘decolonization’ since keywords should ideally be unique terms not found in the title.
Answer: We really appreciate the comment, the keywords were changed.
L 52. High mortality rates
Answer: You are right, it was corrected.
L 56. I suggest considering 'commonly' or 'routinely' as more precise alternatives to 'frequently'.
Answer: Thank you very much for the suggestion, the word was changed.
L 90, 156, 158, 170, 174, 178, 180, 184, 185, 187, 190, 194, 197, 202, 212, 223, 225, 228, 230, 235, 238, 240, 247, 249. C. auris (italics)
Answer: It was written in italics.
L 155. Lavandula angustifolia (italics)
Answer: It was written in italics.
L 185. C. albicans (italics)
Answer: It was written in italics.
L 271. In vitro (italics)
Answer: It was written in italics.
Round 2
Reviewer 2 Report
Comments and Suggestions for Authors
The authors have effectively responded to all my comments and the manuscript has been significantly improved for publication in Tropical Med